# Thyroid Eye Disease Following SARS-CoV-2 Vaccination: Experience of a Case Series

**DOI:** 10.3390/vaccines14010037

**Published:** 2025-12-28

**Authors:** Alin Abreu Lomba, María Elena Tello-Cajiao, Mónica Morales, Alexander Martínez, Mauricio Andrés Salazar Moreno, David Alexander Vernaza Trujillo, Alice Gaibor-Pazmiño, Juan S. Izquierdo-Condoy

**Affiliations:** 1Endocrinology Department, Clínica Imbanaco Grupo QuirónSalud, Cali 760042, Colombia; 2Department of Internal Medicine, Universidad Libre, Cali 760043, Colombia; 3Vision and Eye Health Research Group-VISOC, Universidad del Valle, Cali 760001, Colombia; 4Department of Ophthalmology, Clínica Imbanaco Grupo QuirónSalud, Cali 760042, Colombia; 5Department of Internal Medicine, Geriatrician, Universidad de Caldas, Manizales 170001, Colombia; 6Department of Public Health, Pontificia Universidad Javeriana, Cali 760035, Colombia; 7Clinical Research Center, Clínica Imbanaco, Cali 760042, Colombia; 8One Health Research Group, Universidad de las Américas, Quito 170124, Ecuador

**Keywords:** thyroid eye disease, Graves’ orbitopathy, COVID-19 vaccines, SARS-CoV-2 vaccination, autoimmune thyroid disease

## Abstract

**Background:** Thyroid eye disease (TED), or Graves’ orbitopathy, is the most common extra-thyroidal manifestation of Graves’ disease, but it has only rarely been reported after SARS-CoV-2 vaccination. Autoimmune thyroid disease, including subacute thyroiditis and Graves’ disease, has been described following COVID-19 vaccination; we present a case series of TED occurring shortly after different COVID-19 vaccines to provide clinical data on this potential safety signal. **Case presentation:** We describe five women (mean age 47 years; range 27–69) who developed TED 3–20 days after COVID-19 vaccination with mRNA or adenoviral vector vaccines, three of whom had pre-existing thyroid disease. Presentations included ocular and retro-orbital pain, exophthalmos, headache, goiter, tremor, depressive symptoms, and, in one case, anterior neck pain and fever. TED severity (ETA/EUGOGO) ranged from mild to severe, with frequent findings of suppressed TSH, elevated thyroid autoantibodies, and inflammatory markers, as well as imaging evidence of exophthalmos, extraocular muscle enlargement, and diffuse or multinodular goiter. Management with intravenous corticosteroids, selenium, levothyroxine adjustment, and/or intramuscular corticosteroids led to improvement in thyroid function and inflammation by 3 months, although mild TED often persisted. **Conclusions:** This case series supports a temporal association between COVID-19 vaccination and new-onset or exacerbated TED in individuals with autoimmune thyroid disease. Although vaccination benefits outweigh potential risks, clinicians should remain alert to ocular and thyroid symptoms after immunization to ensure timely diagnosis and management.

## 1. Introduction

Vaccines against severe acute respiratory syndrome coronavirus 2 (SARS-CoV-2) have demonstrated their safety and efficacy in preventing infection and severe disease [1,2]. Currently, four major vaccine platforms are used worldwide: messenger RNA (mRNA) vaccines (Pfizer–BioNTech, Moderna), adenoviral vector vaccines (AstraZeneca, Janssen), inactivated virus vaccines, and protein subunit vaccines. Although mRNA and vector-based vaccines are highly effective, several reports have described autoimmune and thyroid-related complications following immunization, including subacute thyroiditis, Graves’ disease, and, more rarely, thyroid eye disease (TED) [3,4,5,6,7]. Beyond preventing severe acute disease, COVID-19 vaccination has also been associated with a lower burden of long COVID symptoms, while most vaccine-related adverse events are mild and self-limited [8,9].

TED, also known as Graves’ orbitopathy (GO), is the most common extra-thyroidal manifestation of Graves’ disease (GD). It can occur less frequently in patients with chronic autoimmune thyroiditis (CAT) [10] and, more rarely, in euthyroid or hypothyroid patients with CAT [11].

GO is relatively uncommon, with an estimated incidence of 0.54–0.9 cases per 100,000 men per year and 2.67–3.3 cases per 100,000 women per year [10]. Known risk factors include smoking, thyroid dysfunction, and elevated serum levels of thyrotropin receptor antibodies. Most patients have a history of thyroid disease, and the main clinical manifestations include exophthalmos, blurred vision, eyelid swelling, diplopia, and strabismus [12,13].

Since Rubinstein et al. reported the first case of TED after anti–SARS-CoV-2 vaccination in a patient with GD [14], an increasing number of case reports and series have described new diagnoses, recurrences, and relapses of GD following the administration of anti–SARS-CoV-2 vaccines, including presentations with TED [13,14,15,16,17,18].

Although a temporal association between vaccination and thyroid disorders has been observed, a clear causal relationship has not yet been established due to the limited size and quality of the published data [3]. Previous reports describing thyroid eye disease or autoimmune thyroid disorders following SARS-CoV-2 vaccination have been limited to isolated case reports and small case series. Across published reports, onset has most often occurred within the first 1–3 weeks after vaccination and ranged from mild ocular discomfort to moderate to severe orbitopathy [19,20]. Most cases showed partial or complete improvement following standard medical management, including corticosteroids or conservative therapy, although persistent or recurrent symptoms have also been described [21]. With the following five cases of TED after COVID-19 vaccination, we aim to contribute to the understanding of the possible relationship between SARS-CoV-2 vaccination and thyroid eye disease.

## 2. Case Presentation

A series of five cases is presented; all patients were women with a mean age of 47 years (range 27–69 years). TED occurred between 3 and 20 days after receiving a COVID-19 vaccine. These patients were referred to the ophthalmology service, and TED was classified using the European Thyroid Association/European Group on Graves’ Orbitopathy (ETA/EUGOGO) scale (EUGOGO) [22]. Individualized management was provided, and patients were followed for 3 months. Baseline sociodemographic, clinical, laboratory, and imaging characteristics are summarized in Table 1, and representative clinical and radiologic findings are shown in Figure 1.

### 2.1. Case 1

#### 2.1.1. Evaluation and Diagnosis

A 63-year-old woman with a history of primary hypothyroidism, hypertension, unspecified dyslipidemia, type 2 diabetes mellitus, nephrolithiasis, and cholelithiasis, under regular internal medicine follow-up, presented with no previous history of thyroid eye disease or other orbital pathology.

She developed right retro-orbital pain and ipsilateral hemicranial headache 3 days after receiving the third dose of a COVID-19 vaccine (mRNA-1273, Moderna). She had previously received two doses of the AstraZeneca vaccine. Vaccination history was obtained from the patient and confirmed through available medical records.

On physical examination, she was classified as having grade I obesity. Vital signs revealed mildly elevated blood pressure with heart rate within normal limits. Ophthalmologic examination showed bilateral exophthalmos, more marked on the right side. Laboratory tests revealed normal TSH, free T4, free T3, Anti-TPO, and Anti-Tg levels, with elevated ESR and CRP. Orbital computed tomography (CT) demonstrated bilateral exophthalmos with right predominance. The ophthalmology team diagnosed moderate-to-severe TED according to the EUGOGO classification, with exophthalmos < 23 mm and 2 mm of eyelid retraction, without diplopia. Thyroid ultrasound revealed a diffuse hypoechoic goiter.

#### 2.1.2. Therapeutic Intervention

The patient received intravenous methylprednisolone at a dose of 500 mg weekly for 6 doses, followed by 250 mg weekly for 6 additional doses.

#### 2.1.3. Follow-Up

At the 3-month follow-up, the patient was asymptomatic. Thyroid function tests and acute-phase reactants were within normal ranges. Her baseline treatments for comorbidities were continued, and gradual tapering of oral corticosteroids was initiated.

### 2.2. Case 2

#### 2.2.1. Evaluation and Diagnosis

A 29-year-old woman with no significant past medical history and no prior thyroid dysfunction or thyroid eye disease presented 5 days after receiving the third dose of a COVID-19 vaccine (BNT162b2, Pfizer–BioNTech). She had previously received two doses of the AstraZeneca vaccine. She complained of frontal headache, bilateral eye pain, goiter, and upper limb tremors. Vaccination history was obtained from the patient and confirmed through medical records.

Physical examination revealed no significant abnormalities. Laboratory tests showed suppressed TSH, normal free T4, elevated Anti-TPO and Anti-Tg antibodies, and increased TRAb. Ophthalmologic examination demonstrated bilateral exophthalmos. Orbital CT revealed bilateral exophthalmos with increased volume of the medial rectus muscles, consistent with grade 4 exophthalmos and severe TED. Thyroid ultrasound performed by the internal medicine team showed a vascularized diffuse goiter with heterogeneous echogenicity. The right lobe measured 5 cm, the left lobe 4.9 cm, and the isthmus 3 cm.

#### 2.2.2. Therapeutic Intervention

Intravenous corticosteroids were administered for 12 weeks: 500 mg weekly for 6 doses, followed by 250 mg weekly for 6 additional doses. During ophthalmologic follow-up, a 4 mm reduction in exophthalmos volume was documented, with an improvement from severe to moderate TED according to the EUGOGO classification. Subsequently, 1 mL of intramuscular corticosteroid was prescribed monthly for three months.

#### 2.2.3. Follow-Up

At the 3-month follow-up with internal medicine, the patient remained hemodynamically stable and clinically improved. Laboratory tests showed recovery of thyroid function and improvement in acute-phase reactants.

### 2.3. Case 3

#### 2.3.1. Evaluation and Diagnosis

A 47-year-old woman with controlled hypertension, managed by internal medicine, and no previous history of thyroid dysfunction or thyroid eye disease presented 20 days after receiving a single dose of a COVID-19 vaccine (Ad26.COV2.S, Janssen) with mild bilateral exophthalmos. Vaccination history was corroborated with medical records.

Ophthalmologic examination confirmed mild bilateral exophthalmos. Laboratory tests showed normal TSH, low free T4 and total T3, elevated Anti-TPO and Anti-Tg antibodies, normal TRAb, and increased ESR and CRP. Orbital CT demonstrated left-sided exophthalmos with hypertrophy of the ipsilateral oblique and rectus muscles, leading to a diagnosis of moderate-to-severe TED.

#### 2.3.2. Therapeutic Intervention

Management consisted of selenium supplementation. Thyroid ultrasound performed by the internal medicine team revealed a normal-sized gland with sonographic features of chronic thyroiditis.

#### 2.3.3. Follow-Up

At the first 3-month follow-up, TSH was within the normal range, free T4 remained slightly below normal, Anti-TPO was persistently elevated, and Anti-Tg had normalized. The patient continued selenium therapy without significant clinical improvement in exophthalmos.

### 2.4. Case 4

#### 2.4.1. Evaluation and Diagnosis

A 27-year-old woman with a history of primary hypothyroidism, under internal medicine control and on levothyroxine, with no previous thyroid eye disease, presented 18 days after receiving the second dose of a COVID-19 vaccine (BNT162b2, Pfizer–BioNTech). She reported weakness, drowsiness, and low mood. Vaccination history was obtained from the patient and verified through medical records.

Physical examination was unremarkable. Ophthalmologic evaluation identified mild right-sided exophthalmos. Laboratory tests showed low TSH, normal free T4, normal Anti-TPO, Anti-Tg antibodies, and TRAb, with elevated CRP and normal ESR. Orbital CT revealed orbitopathy with enlargement of the rectus muscles and no changes in the oblique muscles, consistent with mild TED.

#### 2.4.2. Therapeutic Intervention

Treatment included selenium supplementation and reduction of the levothyroxine dose. Thyroid ultrasound showed findings compatible with chronic thyroiditis.

#### 2.4.3. Follow-Up

At the 3-month follow-up, thyroid function had normalized, ESR remained unchanged, and CRP was slightly elevated. Selenium and levothyroxine therapy continued due to persistent, although mild, exophthalmos.

### 2.5. Case 5

#### 2.5.1. Evaluation and Diagnosis

A 69-year-old woman with a history of hypothyroidism, previous multinodular goiter, and grade I obesity, without prior thyroid eye disease, presented 19 days after receiving the third dose of a COVID-19 vaccine (BNT162b2, Pfizer–BioNTech). She complained of neck pain, odynophagia, headache, and fever. Vaccination history was derived from patient report and confirmed in the medical record.

Initially, she was diagnosed with tonsillitis in the emergency department and received empirical treatment. Four days after initiating therapy, her pain improved, but she developed mild bilateral exophthalmos. Physical examination revealed obesity, normal vital signs, mild bilateral exophthalmos, and a tender goiter on palpation. Laboratory tests showed low TSH, normal free T4, elevated Anti-TPO and Anti-Tg antibodies, normal TRAb, and increased ESR and CRP. Orbital CT demonstrated bilateral orbitopathy with enlargement of the rectus muscles, consistent with mild bilateral TED. Thyroid ultrasound revealed a multinodular goiter classified as TIRADS 2. Thyroid scintigraphy with Tc-99 showed 1% uptake, consistent with a hypofunctioning goiter and subacute thyroiditis.

#### 2.5.2. Therapeutic Intervention

Management consisted of intramuscular corticosteroids and oral analgesics every 6 h for one month.

#### 2.5.3. Follow-Up

At the 3-month follow-up, TSH and free T4 were slightly decreased, ESR and CRP remained elevated, Anti-Tg levels were within normal limits, and Anti-TPO and TRAb were elevated, with persistent mild exophthalmos.

#### 2.5.4. CAS (Clinical Activity Score)

A retrospective 7-item CAS (0–7) [23] was conservatively estimated from explicitly documented ocular inflammatory signs and symptoms (retrobulbar pain, pain on eye movements, eyelid erythema, conjunctival erythema, eyelid swelling, chemosis, and caruncle/plica inflammation). Because CAS was not prospectively recorded and some items were not systematically documented, missing CAS items were not assumed to be present; therefore, CAS may be underestimated. Based on the available documentation, patients 1, 2, and 5 were classified as having higher inflammatory activity (estimated CAS ≥ 3/7) and received systemic corticosteroid therapy, whereas patients 3 and 4 had lower inflammatory activity (estimated CAS ≤ 2/7) and were managed conservatively with selenium and optimization of thyroid function (Table 2).

## 3. Discussion

This study describes the cases of five women who developed TED within days to weeks after receiving SARS-CoV-2 vaccination. Overall, we observed a spectrum of TED severity from mild to severe, frequent evidence of underlying or newly unmasked AITD, and only partial improvement in some patients despite guideline-based management.

Most cases of TED are mild and non-progressive, whereas moderate-to-severe forms occur in approximately 5–6% of patients. Even mild disease represents a substantial public health burden due to both direct and indirect costs, and moderate-to-severe TED poses a major therapeutic challenge because of its often-incomplete response to available treatments. Therapeutic decisions are based on the clinical presentation, activity, severity, and duration of the disease [15,24].

In our series of five patients, two primarily presented with ocular and retro-orbital pain accompanied by exophthalmos. Additional symptoms across the series included hemicranial and frontal headache, goiter, tremor, depressive symptoms (fatigue, drowsiness, low mood), and, in one patient, anterior neck pain, odynophagia, and fever suggestive of thyroiditis. One patient presented with severe bilateral TED that improved to moderate disease following high-dose intravenous corticosteroid therapy. Another had moderate bilateral TED, and the remaining three had mild TED, one of them with bilateral involvement. The patient with predominant depressive symptoms presented with unilateral exophthalmos. Imaging studies demonstrated bilateral exophthalmos in two patients, unilateral exophthalmos in two patients, and bilateral orbitopathy in one patient. Additional findings included enlargement of the rectus and oblique extraocular muscles.

Thyroid ultrasound in three patients revealed goiter (two diffuse and one multinodular), and in two patients, sonographic findings were consistent with chronic thyroiditis. Biochemically, most patients had suppressed TSH with variable free T4 and T3 levels, and several showed positive Anti-TPO and Anti-Tg antibodies, supporting the diagnosis of AITD. Some patients had positive TRAb, suggesting an underlying Graves’ disease component, whereas others had negative TRAb and a biochemical profile more suggestive of subacute or autoimmune thyroiditis with associated TED. Corticosteroid therapy was reserved for patients with more active or severe TED, whereas selenium supplementation and adjustment of levothyroxine dosing were preferred in cases of milder disease or underlying Graves’ hyperthyroidism.

Management decisions in our case series were guided by disease activity and severity, in accordance with current EUGOGO recommendations [24]. Patients with moderate-to-severe active TED, characterized by higher estimated Clinical Activity Scores (CAS ≥ 3), received systemic corticosteroid therapy, whereas those with mild or less active disease (CAS ≤ 2) were managed conservatively with selenium supplementation and optimization of thyroid function [23]. These differences in clinical severity, inflammatory activity, and therapeutic response raise important questions regarding the immunopathogenic mechanisms that may link SARS-CoV-2 vaccination with the development or exacerbation of thyroid eye disease [25,26].

COVID-19 vaccines induce a strong innate immune response, activating pattern recognition receptors and promoting the release of pro-inflammatory cytokines. In predisposed individuals, this immune activation may facilitate bystander activation of autoreactive lymphocytes, contributing to the loss of tolerance against thyroid and orbital antigens. Molecular mimicry between SARS-CoV-2 spike protein epitopes and self-antigens has also been proposed as a potential mechanism underlying vaccine-associated autoimmune thyroid disorders [20,25,27]. Although mRNA vaccines do not contain classical adjuvants, lipid nanoparticles exhibit adjuvant-like properties, enhancing antigen presentation and immune stimulation. Viral vector vaccines similarly activate innate immune pathways. This heightened immune activation may promote orbital fibroblast stimulation via TSHR and IGF-1R signaling, a central mechanism in TED pathophysiology [28].

Importantly, in our series, patients with higher estimated inflammatory activity (CAS ≥ 3) exhibited a more aggressive clinical presentation and required systemic corticosteroid therapy, with partial to significant improvement. In contrast, patients with lower estimated CAS (≤2) showed milder disease and persistent but stable symptoms under conservative management with selenium. This clinical pattern supports the concept that vaccine-related immune activation may preferentially unmask or exacerbate active inflammatory disease rather than induce de novo severe TED.

As reported in the literature, autoimmune thyroid disease (AITD) can develop or worsen after receiving the COVID-19 vaccine, in patients with or without a history of thyroid disease [29,30]. In our report, three out of five patients had a history of thyroid disease: two with hypothyroidism treated with levothyroxine and one with a history of multinodular goiter. At the time of consultation, two patients were euthyroid, one had hyperthyroidism, and one had overtreated hypothyroidism requiring a reduction in levothyroxine dose. Three patients had elevated Anti-TPO levels, one had elevated Anti-Tg levels, and three had elevated TRAb, further supporting AITD [31]. Four patients had elevated CRP levels, and three had elevated ESR levels. These findings suggest that vaccination may act as an immune trigger capable of unmasking or exacerbating pre-existing thyroid autoimmunity rather than inducing de novo disease.

These findings should be interpreted with caution, as they do not negate the substantial public health impact of COVID-19 vaccination. Large cohort studies have shown that most vaccine-related adverse events are mild or moderate and that serious reactions are uncommon [32]. In contrast, absence or incompleteness of vaccination has been associated with a higher burden of long COVID, encompassing a wide spectrum of persistent symptoms, and with an increased risk of acute cardiovascular events such as myocardial infarction [8,33,34].

Finally, while ocular inflammatory conditions such as uveitis, scleritis, and posterior neuritis have been reported following administration of the COVID-19 vaccine, Graves’ orbitopathy associated with immunization against the same pathogen has not yet been widely considered among the recognized adverse effects. In our report, TED occurred in temporal association with vaccination. Despite appropriate treatment, complete improvement was not achieved in all patients after the 3-month follow-up. Differences between vaccine platforms may influence the risk of post-vaccination autoimmune reactions. mRNA vaccines have been more frequently associated with autoimmune thyroid events, possibly due to robust innate immune activation and adjuvant-like effects mediated by lipid nanoparticles that deliver the mRNA sequence [29,31]. Adenoviral vector vaccines have also been implicated, although less frequently, suggesting that immune dysregulation rather than a single specific vaccine component may underlie these phenomena [28]. However, currently available data remains limited, and a causal relationship has not been established.

Limitations

This study has several limitations inherent to its design. As a descriptive case series, it does not allow estimation of incidence, risk, or causal relationships between SARS-CoV-2 vaccination and TED. Background rates of TED in the source population were not available; therefore, a coincidental temporal association cannot be excluded. Accordingly, our findings should be interpreted as hypothesis-generating rather than confirmation.

Complete pre-vaccination thyroid and ophthalmologic data were not available for all patients. In our series, three patients had a documented history of thyroid disease, whereas two had no known prior thyroid disorder. Importantly, one patient with multinodular goiter had documented negative anti-thyroid antibodies several years before vaccination. These findings suggest that SARS-CoV-2 vaccination may have acted as an immune trigger capable of unmasking previously subclinical disease or accelerating an underlying autoimmune process rather than uniformly inducing de novo pathology; however, this hypothesis cannot be formally tested within the constraints of the present study.

Finally, the small sample size and short follow-up period limit the generalizability of our findings and our ability to assess long-term outcomes. Larger, controlled studies with systematic baseline and longitudinal follow-up are needed to better characterize the relationship between SARS-CoV-2 vaccination and TED.

## 4. Conclusions

An increasing number of cases of thyroid disorders, including TED, have been reported in association with SARS-CoV-2 infection and, more recently, following immunization with different COVID-19 vaccines. In our case series, TED developed within days to weeks after vaccination across different vaccine platforms, including predominantly mRNA- and adenoviral vector-based vaccines, affecting women and frequently occurring in the context of underlying or newly unmasked autoimmune thyroid disease.

Although the benefits of COVID-19 vaccination clearly outweigh the risks associated with infection, healthcare professionals should remain aware of potential autoimmune-related manifestations following vaccination. Early recognition of symptoms suggestive of TED or thyroid dysfunction is essential to allow timely diagnosis and initiation of appropriate management, with the aim of improving prognosis and quality of life. Further research in larger, controlled studies is warranted to better clarify the relationship between different vaccine platforms, immune activation, and autoimmune thyroid disease. Our observations should be interpreted as hypothesis-generating rather than causal.

## Figures and Tables

**Figure 1 vaccines-14-00037-f001:**
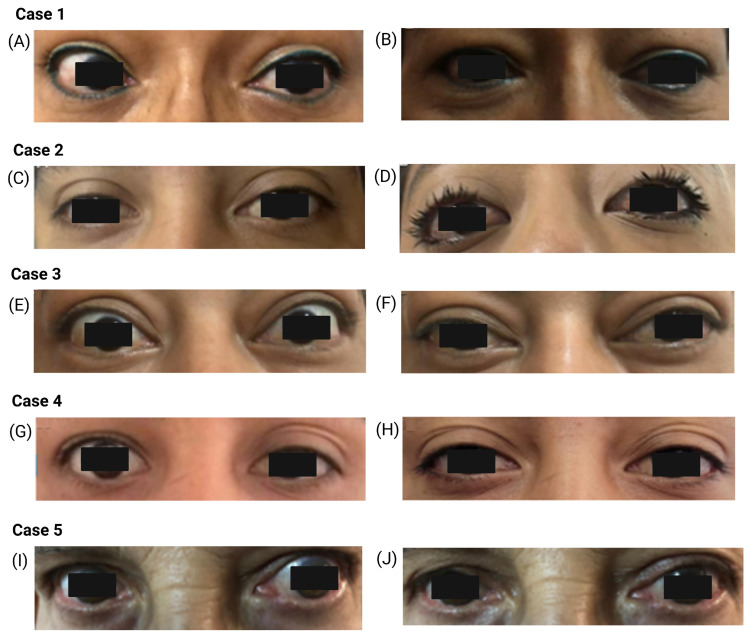
Clinical photographs of patients with thyroid eye disease after COVID-19 vaccination. Images at presentation (**A**,**C**,**E**,**G**,**I**) and at 3-month follow-up after treatment (**B**,**D**,**F**,**H**,**J**) are shown for Case 1–5. The images illustrate upper eyelid retraction, periorbital edema, and exophthalmos at baseline, with partial improvement after corticosteroid or selenium therapy. Black rectangles were added to protect patient identity.

**Table 1 vaccines-14-00037-t001:** Baseline sociodemographic, clinical, laboratory, imaging, treatment, and outcome characteristics of patients with thyroid eye disease after COVID-19 vaccination.

	Case 1	Case 2	Case 3	Case 4	Case 5	Reference Values/Units
**Demographics and vaccination**						
Age	63	29	47	27	69	years
Sex	Female	Female	Female	Female	Female	–
Total COVID-19 vaccine doses before symptom onset	3	3	1	2	3	doses
Vaccine regimen	ChAdOx1 nCoV-19 × 2; mRNA-1273 (3rd dose)	ChAdOx1 nCoV-19 × 2; BNT162b2 (3rd dose)	Ad26.COV2.S (single dose)	BNT162b2 × 2	BNT162b2 × 3	–
Days from last vaccine dose to symptom onset	3	5	20	18	19	days
**Clinical presentation and past medical history**						
Main presenting symptoms	Right retro-orbital pain; ipsilateral hemicranial headache	Bilateral eye pain; frontal headache; goiter; tremor	Mild bilateral exophthalmos	Fatigue; drowsiness; low mood	Anterior neck pain; odynophagia; headache; fever (39 °C)	–
Exophthalmos at presentation	Bilateral, right-sided predominance	Bilateral	Unilateral (left eye, mild)	Unilateral (right eye, mild)	Bilateral (mild)	–
Past medical history	Primary hypothyroidism; hypertension; dyslipidemia; type 2 diabetes mellitus; nephrolithiasis; cholelithiasis	None	Hypertension	Primary hypothyroidism, well controlled for 10 years	Previous multinodular goiter (negative Anti-TPO and Anti-Tg in 2019); obesity grade I	–
Chronic medication	Levothyroxine 50 µg/day; losartan	None	Telmisartan 80 mg/day	Levothyroxine 88 µg/day	None	–
**Anthropometrics and vital signs**						
Weight	82	67	62	69	83	kg
Height	1.56	1.65	1.54	1.70	1.62	m
BMI	33.7	24.6	26.1	23.9	31.6	kg/m^2^
Blood pressure	140/80	112/76	130/71	130/70	126/82	≈120/80 mmHg
Heart rate	86	68	86	98	96	60–80 bpm
**Thyroid function tests**						
TSH	1.71	0.12	0.80	0.20	0.10	0.54–4.07 µIU/mL
Free T4	1.38	1.50	1.16	1.71	2.10	0.92–1.53 ng/dL
Free T3	2.10	–	–	–	–	2.0–4.4 pg/mL
Total T3	–	–	80	–	–	(laboratory-specific range, ng/dL)
**Thyroid autoantibodies and inflammatory markers**						
Anti-TPO	64.0	1.8	276	12.0	576	<35 IU/mL
Anti-Tg	18.0	14.0	67	4.0	6.0	<40 IU/mL
TRAb	6.8	6.0	0.3	0.4	8.0	<1.75 U/L
ESR	10	69	82	70	136	0–20 mm/h
CRP	18	14	26	14	18	<5 mg/dL
Imaging						
Orbital CT	Bilateral exophthalmos with right-sided predominance	Bilateral exophthalmos with increased volume of the medial rectus muscles	Left-sided exophthalmos with hypertrophy of left oblique and rectus muscles	Orbitopathy with enlargement of rectus muscles; oblique muscles unchanged	Bilateral thyroid orbitopathy with enlargement of rectus muscles in both orbits	–
Thyroid ultrasound	Diffuse hypoechoic goiter	Diffusely vascularized goiter on Doppler; heterogeneous echogenicity (right lobe 5 cm, left lobe 4.9 cm, isthmus 3 cm)	Normal-sized gland with features of thyroiditis	Findings consistent with chronic thyroiditis	Multinodular goiter (TI-RADS 2)	–
Thyroid scintigraphy (Tc-99)	–	–	–	–	Goiter with 1% uptake on Tc-99, consistent with hypofunctioning goiter and subacute thyroiditis	–
**Ophthalmologic assessment, treatment, and outcome**						
Ophthalmologic assessment	Moderate Graves’ orbitopathy with soft-tissue involvement; right-dominant exophthalmos < 23 mm; 2 mm eyelid retraction; no diplopia	Severe TED with grade 4 exophthalmos	Mild TED	Mild TED, right eye	Mild bilateral TED	–
Treatment	Intravenous methylprednisolone 500 mg weekly × 6, then 250 mg weekly × 6	Intravenous methylprednisolone 500 mg weekly × 6, then 250 mg weekly × 6	Selenium 200 µg/day	Selenium 200 µg/day; levothyroxine dose reduced after 2 months	Betamethasone (Diprofos) 7 mg IM	–
Short-term outcome	Improved; asymptomatic at 4 months; thyroid function and inflammatory markers normalized; tapering of oral prednisone to 5 mg/day	Improved; 4 mm reduction in exophthalmos; TED improved from severe to moderate	Thyroid function normalized at 3 months; Anti-TPO remained elevated but decreasing; exophthalmos persisted; continued selenium	Improved; thyroid function and inflammatory markers normal; mild exophthalmos persisted; continued selenium and levothyroxine	Partial improvement at 4 months with persistent mild bilateral exophthalmos and elevated Anti-TPO; at 1-year follow-up asymptomatic with normal thyroid function and antibodies; ultrasound showed small nodular goiter with chronic thyroiditis	–

**Abbreviations:** Anti-Tg, anti-thyroglobulin antibodies; Anti-TPO, anti-thyroid peroxidase antibodies; BMI, body mass index; CRP, C-reactive protein; CT, computed tomography; ESR, erythrocyte sedimentation rate; IM, intramuscular; LE, left eye; RE, right eye; Tc-99, technetium-99 m; TED, thyroid eye disease; TRAb, thyroid-stimulating hormone receptor antibodies; TSH, thyroid-stimulating hormone.

**Table 2 vaccines-14-00037-t002:** Retrospective Clinical Activity Score (CAS), treatment approach, and short-term outcomes in patients with thyroid eye disease following SARS-CoV-2 vaccination.

Estimated CAS (0–7)	Patients	CAS	Treatment	Outcome
CAS ≥ 3	1, 2, 5	Active	Corticosteroids	Partial to significant improvement
CAS ≤ 2	3, 4	Low activity	Selenium	Persistent mild disease

## Data Availability

The original contributions presented in this study are included in the article. Further inquiries can be directed to the corresponding author.

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
