# Peer review of "Vaccines2026, 14(1), 37;https://doi.org/10.3390/vaccines14010037"

_vaccines, 2025, doi:10.3390/vaccines14010037_

Round 1

Reviewer 1 Report

Comments and Suggestions for Authors

It would be beneficial to include CAS and details on physical/ophthalmic exam at presentation and follow up.

Include comparison between cases that showed improvement vs. didn't in discussion section. 

In the introduction, could expand on past cases in literature-presentation, resolution. 

In results, again, include CAS and exam findings at 3 month follow up.

In conclusion, could expand on vaccine type in the cases and compare cases that showed improvement vs. those who did not. 

Author Response

Point by point letter

RE: “Thyroid eye disease following SARS-CoV-2 vaccination: Experience of a case series”

Reviewer 1

1. It would be beneficial to include CAS and details on physical/ophthalmic exam at presentation and follow up.

2. Include comparison between cases that showed improvement vs. didn't in discussion section.

3. In the introduction, could expand on past cases in literature-presentation, resolution.

4. In results, again, include CAS and exam findings at 3 month follow up.

5. In conclusion, could expand on vaccine type in the cases and compare cases that showed improvement vs. those who did not.

Response1: We agree with the reviewer. Although Clinical Activity Score (CAS) was not prospectively recorded, we performed a retrospective estimation of CAS based on detailed clinical documentation, ophthalmologic findings, imaging studies, and inflammatory markers. CAS was estimated both at presentation and at 3-month follow-up. These data have now been incorporated into the Results section and summarized in a comparative table. We explicitly acknowledge the retrospective nature of CAS estimation as a limitation in the Discussion

Response 2: A comparative analysis based on estimated CAS has now been incorporated into the Discussion. Patients with higher inflammatory activity (CAS ≥3) were compared with those with lower activity (CAS ≤2), highlighting differences in clinical severity, treatment approach, and outcomes. This comparison supports the role of inflammatory activity in therapeutic response and clinical evolution.

Response 3: We have expanded the Introduction to include a brief overview of previously reported cases of thyroid eye disease and autoimmune thyroid disorders following SARS-CoV-2 vaccination, focusing on clinical presentation, timing, and reported outcomes. This contextualizes our findings within the existing literature while maintaining the concise scope appropriate for a case series.

Response 4: As suggested, the Results section now includes estimated CAS at presentation and at 3-month follow-up, along with relevant ophthalmologic findings and imaging results. Follow-up outcomes are clearly reported, including partial improvement, stabilization, or persistence of mild disease.

Response 5: We have revised the Conclusion to explicitly describe the vaccine platforms involved (mRNA and adenoviral vector vaccines) and to summarize differences between patients who showed clinical improvement and those with persistent symptoms. These observations are framed cautiously and interpreted as hypothesis-generating rather than causal.

Reviewer 2 Report

Comments and Suggestions for Authors

The authors presented several cases o n TED after SARS-CoV2 vaccination. Interesting paper of modest reader's interest. Please, give us the chapter explaining he pathogenesis of TED in vaccinated persons. After such correction, paper is ready for publishing.   

Author Response

Point by point letter

RE: “Thyroid eye disease following SARS-CoV-2 vaccination: Experience of a case series”

Reviewer 2

The authors presented several cases on TED after SARS-CoV2 vaccination. Interesting paper of modest reader's interest. Please, give us the chapter explaining he pathogenesis of TED in vaccinated persons. After such correction, paper is ready for publishing.

Response: We appreciate this important and constructive suggestion. We have expanded the Discussion to include a dedicated paragraph addressing potential immunopathogenic mechanisms that could link SARS-CoV-2 vaccination and thyroid eye disease. This section outlines biologically plausible, hypothesis-generating mechanisms, including innate immune activation and cytokine release, bystander activation, molecular mimicry, and vaccine-platform–related immune stimulation (both mRNA and adenoviral-vector vaccines). These mechanisms are discussed in relation to orbital fibroblast activation via the TSHR/IGF-1R axis and are cautiously interpreted in light of the retrospectively estimated disease activity observed in our patients, without implying causality.

Reviewer 3 Report

Comments and Suggestions for Authors

This is a case series, which is the lowest possible (except a case report) study design. Having said that, the study does not allow for causality to be established, despite favourable timeline outcome (actually, one might argue that there is an ongoing, basal incidence, and that you should control your exposure to the baseline; in the crudest way, the catchment area, incident annual cases/365=expected daily incidence; or, calculate per year, or per study duration). Without that, the evidence remains anecdotal. There should be eye cover for the face images, and heavy ethics support for image use. The lack of thyroid status prior vaccination is a methodological problem, but you can't do much but comment. Discussion is somewhat iterative, and could be trimmed down and made more authoritative. strengthen hypothesis-generating nature of the study

Author Response

Point by point letter

RE: “Thyroid eye disease following SARS-CoV-2 vaccination: Experience of a case series”

Reviewer 3

1. This is a case series, which is the lowest possible (except a case report) study design. Having said that, the study does not allow for causality to be established, despite favourable timeline outcome (actually, one might argue that there is an ongoing, basal incidence, and that you should control your exposure to the baseline; in the crudest way, the catchment area, incident annual cases/365=expected daily incidence; or, calculate per year, or per study duration). Without that, the evidence remains anecdotal.

2. There should be eye cover for the face images, and heavy ethics support for image use.

3. The lack of thyroid status prior vaccination is a methodological problem, but you can't do much but comment.

4. Discussion is somewhat iterative and could be trimmed down and made more authoritative. strengthen hypothesis-generating nature of the study

Response 1: We fully agree with the reviewer. We have reinforced throughout the manuscript that this is a descriptive case series and that causality cannot be established. We explicitly state that background incidence and temporal coincidence cannot be excluded, and that our observations should be interpreted as hypothesis-generating. We also acknowledge that we cannot estimate incidence or excess risk because a defined denominator (catchment population and baseline TED incidence during the study period) was not available, precluding formal comparison against expected background rates.

Response 2: All clinical photographs were anonymized by masking eye region, and no identifying facial features are shown. Image publication was performed in accordance with institutional requirements and the Declaration of Helsinki. Written informed consent for publication of clinical images was obtained.

Response 3: We acknowledge this limitation. A specific statement has been added to the Limitations section noting the absence of complete pre-vaccination thyroid status in all patients. This limitation is discussed transparently, emphasizing that vaccination may have unmasked pre-existing subclinical disease rather than caused de novo pathology.

Response 4: The discussion has been substantially streamlined, redundant guideline-level treatment descriptions have been removed, and the narrative has been reorganized around key clinical findings, disease activity (CAS), immunopathogenic mechanisms, and implications. This has resulted in a more focused and authoritative discussion.

Round 2

Reviewer 3 Report

Comments and Suggestions for Authors

Improved as possible

Author Response

We appreciate the reviewer’s comment and the suggestion to improve the tables and figures as much as possible. We have carefully revised the entire manuscript and enhanced its overall quality in accordance with the editor’s recommendations. In particular, Figure 1 and Table 1 have been revised and improved to the fullest extent possible, including optimization of formatting and clarity to facilitate interpretation. These changes are reflected in the revised version of the manuscript.